# Hebbian Learning through the Lens of Sparse Autoencoders

**Nikita Kurdiukov**[*]
T-Tech
krdkvnkt@gmail.com

**Anton Razzhigaev**
FusionBrain Lab
anton.razzhigaev@gmail.com

## Abstract

We establish a theoretical and empirical connection between Hebbian Winner-Take-All (WTA) learning with anti-Hebbian updates and tied-weight sparse autoencoders (SAEs), offering a framework to explain the high selectivity of neurons to patterns induced by biologically inspired learning rules. By training a SAE on token embeddings of a small language model using a gradient-free Hebbian WTA rule with competitive anti-Hebbian plasticity, we demonstrate that such methods implicitly optimize SAE objectives. However, they underperform backpropagation SAEs in reconstruction due to gradient approximations. Hebbian updates approximate reconstruction error (MSE) minimization under tied weights, while anti-Hebbian updates enforce sparsity/feature orthogonality, akin to explicit $L_1/L_2$ penalties in standard SAEs. This alignment with the superposition hypothesis (Elhage et al., 2022) reveals how Hebbian rules disentangle features in overcomplete latent spaces, marking the first application of Hebbian learning to SAEs for language model interpretability.

## 1 Introduction

Despite the long history of biologically inspired learning mechanisms, Hebbian WTA training augmented with anti-Hebbian (negative) updates lacks a comprehendible explanation and a connection to modern machine learning (Földiák, 1990; Pehlevan & Chklovskii, 2014). These rules are notable for their rapid convergence, adversarial robustness, and high selectivity of neurons to input patterns (Krotov & Hopfield, 2019; Journé et al., 2023). While prior work, such as SoftHebb (Moraitis et al., 2022), finds explanation in Bayesian inference, the underlying principles remain partially opaque. As shown in Krotov & Hopfield (2019), the weights of a fully-connected layer trained with their algorithm stores an image prototype or exactly the input images in its weight vectors. We conjecture that this neuronal selectivity is analogous to the feature selectivity exhibited by SAEs used for language model (LM) mechanistic interpretability research (Bricken et al., 2023; Yun et al., 2023; Gao et al., 2024).

**We propose a novel perspective: framing Hebbian WTA learning with anti-Hebbian updates as a form of tied-weight SAE training** (Cunningham et al., 2023). This connection not only demystifies the mechanism behind neuronal selectivity, but also aligns with the superposition hypothesis, which posits that neural networks encode more features than their number of neurons through interfering representations (Olah et al., 2020; Elhage et al., 2022). We shed light onto how Hebbian rules approximate SAE objectives, thereby explaining their empirical advantages in interpretability and efficiency. To validate our hypothesis, we conduct a comparison of SAEs trained via Hebbian WTA rules and standard backpropagation. Our experiments demonstrate that Hebbian SAEs (HSAEs) minimize reconstruction (MSE) and sparsity ($L_1$) objectives akin to traditional SAEs, albeit with a performance gap due to the approximation of gradients by Hebbian updates. Despite this limitation, manual inspection reveals that HSAEs learn monosemantic latents (bottleneck neurons), activating on distinct interpretable input patterns/concepts, mirror-

---

[*]Work done while at FusionBrain Lab.

ing the interpretability benefits of conventional SAEs. Furthermore, we highlight theoretical connections between Hebbian WTA updates and the anti-gradients of SAE objectives under the tied-weight constraint, revealing how anti-Hebbian terms enforce sparsity and/or feature orthogonality, while Hebbian terms drive reconstruction. These insights advance the understanding of biologically plausible learning rules and connects them to the on-going research in mechanistic interpretability and dictionary learning.

## 2 Background

### 2.1 Hebbian Learning and Principal Component Analysis

Hebbian learning (Hebb, 2005) is a learning principle that is often summarized as "neurons that fire together wire together". Unlike backpropagation, which requires global error signals, Hebbian learning uses only local information: synaptic weights of neurons are adjusted based on the correlation of pre- and post-synaptic activity, i.e. inputs and outputs of a layer. In terms of machine learning, Hebbian rules are unsupervised representation learning methods that aim to extract useful representations of the data for downstream tasks. The simplest Hebbian rule is $\Delta \boldsymbol{w} = \boldsymbol{y}\boldsymbol{x}^T$, but it makes the weight grow unboundedly. To solve that issue the well-known Oja's rule (Oja, 1982) incorporates the weight vector normalization into the update rule, resulting in the algorithm to extract the first principal component of the data. However, applying the Oja's rule for more neurons in a layer is not sufficient for finding later components, since there's no interaction between neurons. To handle this, methods like Sanger's rule and (Oja's) subspace algorithm were introduced (Sanger, 1989; Oja, 1989). The table with expressions of learning rules can be found in Table A.1. In such a way, Hebbian learning has a tight connection to online PCA, formulated in an algorithmic fashion, i.e. through weight updates. However, it can be rewritten in an optimization way, e.g. PCA can be done with a linear reconstructing autoencoder (AE) (Plaut, 2018). Furthermore, Hebbian learning can be performed in the modern deep learning framework, PyTorch, by formulating weight updates in terms of loss functions (Miconi, 2021).

Hebbian Winner-Take-All (WTA) learning refers to a competitive learning approach, where weight updates follow Hebbian rules, but only the most active neuron(s) receive Hebbian (positive) updates, while other neurons may receive anti-Hebbian (negative) updates. For example, only the top-1 neuron, by its output value in a layer, updates its weight vector positively, or the top-5 neuron gets a negative update, using a sign switch. This competitive mechanism enforces the selectivity of neurons to patterns (Moraitis et al., 2022).

### 2.2 Sparse Dictionary Learning

Sparse dictionary learning aims to represent input data in terms of a sparse linear combination of basic dictionary elements (Olshausen & Field, 1997). Sparse autoencoder is one of such methods, which is widely used in LM interpretability research to learn a dictionary of directions in activation space (Bricken et al., 2023). It is trained to reconstruct activations of another neural network, e.g. LM, through a bottleneck layer that is constrained to be *sparse* (typically via an $L_1$ penalty on bottleneck acivations or a $k$-sparse constraint). This sparsity prior encourages the network to discover a distributed representation where each unit fires only for a limited set of patterns.

### 2.3 SAE for Interpretability of Language Models

According to the superposition hypothesis language models tend to store more features inside of their representations then they have dimensions. This is possible by representing a feature with a linear combination of neurons, and due to the sparse activation of features themselves. SAE takes these features out of superposition by bringing them into the overcomplete basis, the bottleneck layer, where neurons can align with features one-to-one. Throughout this paper, we refer to neurons in the bottleneck of an SAE as *latents*, akin to Lieberum et al. (2024).

## 3 Methodology

In our experiments, we employ a sparse autoencoder (SAE) of the form:

$$\boldsymbol{y} = \text{ReLU}(\boldsymbol{W}_{\text{enc}}\boldsymbol{x} + \boldsymbol{b}_{\text{enc}}), \quad \hat{\boldsymbol{x}} = \boldsymbol{W}_{\text{dec}}\,\boldsymbol{y}, \tag{1}$$

with $\boldsymbol{W}_{\text{enc}} \in \mathbb{R}^{K \times d}$, $\boldsymbol{W}_{\text{dec}} \in \mathbb{R}^{d \times K}$. The loss function is:

$$\mathcal{L} \;=\; \|\boldsymbol{x} - \hat{\boldsymbol{x}}\|_2^2 \;+\; \alpha\,\|\boldsymbol{y}\|_1, \tag{2}$$

where $\alpha$ is a hyperparameter that controls the strength of $L_1$ penalty on the encoder output. A tied-weight SAE means $\boldsymbol{W}_{\text{enc}} = \boldsymbol{W}_{\text{dec}}^\top = \boldsymbol{W}$. We adapt a public SAE codebase[1] and use a ReLU encoder, zeroed decoder bias, and no orthonormal initialization on $\boldsymbol{W}_{\text{dec}}$.

We integrate SoftHebb learning rule from Journé et al. (2023) for the encoder's weights and bias (`HebbSoftKrotovLinear`[2]). During a forward pass, the layer still computes a linear transformation with ReLU activation applied, but weight updates follow the local Hebbian rule, with Softmax activation function, instead of backpropagation. In SAE setups without tied weights, the decoder matrix is trained via backpropagation. Otherwise, the decoding is made with $\boldsymbol{W}_{\text{enc}}^\top$. If a layer uses backpropagation, its Hebbian update is disabled; if it uses Hebbian learning, its parameters are frozen for backpropagation updates.

We evaluate the mean-squared error ($\|\boldsymbol{x} - \hat{\boldsymbol{x}}\|_2^2$), $L_1$ sparsity ($\|\boldsymbol{y}\|_1$), explained variance ratio (EVR) ($1 - \frac{\|\boldsymbol{x} - \hat{\boldsymbol{x}}\|_2^2}{\|\boldsymbol{x} - \bar{\boldsymbol{x}}\|_2^2}$), average $L_0$ norm of activations ($\|\boldsymbol{y}\|_0$), and effective dictionary size (number of latents that activated at least once on the whole batch). The EVR quantifies how much better the model's prediction (reconstruction) $\hat{\boldsymbol{x}}$ compared to the baseline of always predicting mean dense activations $\bar{\boldsymbol{x}}$. EVR close to 1 indicates near-perfect reconstruction, whereas a negative value implies performance worse than predicting the mean. We compute metrics and losses on the same subset used for training.

## 4 Experimental Setup

For our experiments, we use the TinyStories dataset and extract LM activations from the `TinyStories-1M` Transformer-based autoregressive language model, which is trained on synthetic toy data (Eldan & Li, 2023). The token representations are gathered using `TransformerLens`[3] library from the model position `blocks.4.hook_resid_pre`, i.e. from the residual stream. Each sample contains 512 embeddings of dimension 64, yielding 400 K samples (204.8 M embeddings total).

The SAE hyperparameters are $d = 64$, $K = 3200$ and $\alpha = 10^{-2}$. We train for 5 epochs with batch size 16384, repeating each setup under 5 seeds. All weight matrices follow the normal initialization from Journé et al. (2023). The rest of the hyperparameters for the Hebbian learning can be found in Appendix A.2. When backpropagation is applied, we use Adam with a learning rate of $10^{-2}$, gradient norm clipping at 1.0, and $\alpha = 10^{-2}$ for sparsity. Additionally, we conduct an ablation study in which the default PyTorch initialization is used in place of the normal initialization scheme. Default PyTorch initialization uses parametrized Kaiming uniform distribution on the weights and uniform on the bias.

We compare the following setups:

1. **BP**: Both encoder and decoder trained with standard backprop.
2. **BP Tied**: Same as BP, but with one trainable matrix $\boldsymbol{W}_{\text{enc}} = \boldsymbol{W}_{\text{dec}}^\top = \boldsymbol{W}$.
3. **Hebbian Encoder + BP Decoder (BP Dec)**: Encoder weights updated by SoftHebb, decoder - by backprop.
4. **Full Hebbian Tied (Hebb Tied)**: Encoder trained via SoftHebb, decoder is $\boldsymbol{W}_{\text{enc}}^\top$ (no backpropagation-based training).

---

[1] https://github.com/ApolloResearch/e2e_sae
[2] https://github.com/NeuromorphicComputing/SoftHebb
[3] https://github.com/TransformerLensOrg/TransformerLens

Table 1: Evaluation metrics for each training setup. We report the mean (standard deviation) over 5 runs. MSE is the mean-squared error of reconstruction; EVR is the explained variance ratio; $L_1$ is the average sparsity loss; $L_0$ is the average count of active latent units; Dictionary Size is the total number of latent units that become active at least once.

| Setup | MSE | EVR | $L_1$ | $L_0$ | Dict Size |
|---|---|---|---|---|---|
| BP | $4.0 \times 10^{-5} \, (\pm 7.3 \times 10^{-6})$ | $0.929 \, (\pm 0.012)$ | $2.56 \times 10^{-3} \, (\pm 8.9 \times 10^{-5})$ | $32.4 \, (\pm 2.08)$ | $6.12 \times 10^2 \, (\pm 17.5)$ |
| BP Tied | $2.04 \times 10^{-4} \, (\pm 4.3 \times 10^{-5})$ | $0.391 \, (\pm 0.271)$ | $7.04 \times 10^{-3} \, (\pm 8.5 \times 10^{-4})$ | $12.0 \, (\pm 2.40)$ | $2.42 \times 10^1 \, (\pm 5.7)$ |
| BP Dec | $1.63 \times 10^{-4} \, (\pm 5.8 \times 10^{-6})$ | $0.615 \, (\pm 0.025)$ | $1.17 \times 10^{-2} \, (\pm 7.6 \times 10^{-4})$ | $22.3 \, (\pm 1.96)$ | $1.89 \times 10^3 \, (\pm 53.8)$ |
| Hebb Tied | $7.29 \times 10^{-3} \, (\pm 1.75 \times 10^{-3})$ | $-0.520 \, (\pm 0.110)$ | $1.17 \times 10^{-2} \, (\pm 7.6 \times 10^{-4})$ | $22.3 \, (\pm 1.96)$ | $1.89 \times 10^3 \, (\pm 53.8)$ |

## 5 Results

Table 1 summarizes our quantitative results on MSE reconstruction, explained variance (EVR), sparsity loss, samplewise $L_0$ norm and effective dictionary size. As expected, the fully backpropagation-trained baseline (BP) achieves the lowest MSE and sparsity loss. By contrast, our Hebbian-based approach (Hebb Tied) yields a higher reconstruction error but preserves richer dictionary of active units. Interestingly, the hybrid approach with a Hebbian encoder and backprop decoder (BP Dec) provides a middle ground, achieving moderate reconstruction performance alongside reasonably high sparsity. These results illustrate a trade-off between classical backpropagation performance and the interpretability or biological plausibility conferred by Hebbian updates.

Figure 1 shows losses and EVR during training. For clarity, BP Tied's high-variance regions (e.g., negative EVR, std > 1) are omitted. BP setup achieves the lowest MSE (Fig. 1a), aligning with its direct MSE optimization, while Hebbian setups (e.g., Hebb Tied) show higher errors due to gradient approximations. In Fig. 1b, BP maintains the greatest EVR ($\approx 93\%$), BP Dec achieves moderate EVR ($\approx 61.5\%$) , and Hebb Tied's EVR drops below zero (linked to high variance; see Fig. 3 in Appendix). BP Tied hugely underperforms BP due to initialization sensitivity: switching to PyTorch's default initialization improves BP Tied's EVR (>90%), but destabilizes Hebbian setups (NaN outputs). The weight initialization ablation results can be seen in Appendix A.4.

Figure 1c depicts $L_1$ sparsity loss during training. The BP setup achieves the lowest $L_1$ loss, but gradually increases post-MSE convergence. Hebbian setups overlap on the figure due to shared encoder training. They exhibit the greatest $L_1$ loss value, while BP Tied shows comparable $L_1$ loss with high variance. Figure 2 shows sparsity statistics. BP Dec and Hebb Tied setups overlap due to shared encoder training. The left panel reveals the average $L_0$ norm (active units per sample): BP and Hebbian setups initially activate similar counts, but BP's $L_0/L_1$ norms grow post-MSE convergence. BP Tied exhibits the sparsest activations. The right panel highlights effective dictionary size: BP Tied collapses to near-zero rapidly (incompatible with normal initialization), while Hebbian setups activate more dictionary entries, likely due to slower convergence and no implicit $L_1$ minimization.

We provide examples of latent's activations on token embeddings, which are acquired with the same model our SAEs were trained on. You can see the examples in Table 2 for Hebb Tied setup and in Table 3 for BP setup. The manual inspection of the latents' activations reveals the presence of **monosemantic interpretable concepts** for the corner case setups, BP and Hebb Tied. The dataset used for this evaluation is `NeelNanda/pile-10k`, which is different from the training one.

## 6 Discussion

We believe the explanation of pattern selectivity and SAE objectives minimization of Hebbian WTA learning with anti-Hebbian updates is twofold. The Hebbian, positive, weight updates are the truncated anti-gradients of the MSE loss function of the reconstructing tied weight AE (see Table A.1). Since the SoftHebb (Journé et al., 2023) is based on the Oja's rule, the difference from the default MSE gradients are the absence of $\boldsymbol{W}_{dec}$ gradient term and the diagonalization of the autocorrelation matrix of $\boldsymbol{y}$. The anti-Hebbian, negative,

Table 2: Examples of top samples and their contexts that activate latents in Hebbian tied weight SAE, Hebb Tied setup. The activating tokens are highlighted in orange, and the color opacity indicates activation strength.

| Latent index | Pattern | The most latent-activating examples (strength descending order) |
|---|---|---|
| 27 | Token "edd" | \nOverlooking the Sen**edd** from the glassed-
A mile east of B**edd**gelert, this mine
the first ever eist**edd**fod in 1176 |
| 19 | Token "tr" | arrived in the gold-**tr**immed Irish State Coach
or any other non-**tr**usted user) does not
we implement full ray-**tr**acing to calculate $\k |
| 9 | Closing ">" before "</value" | wiki/Astrovirus**>** (accessed July 2019
</value**>**\n
.</string> </value**>**\n |
| 86 | Context of religious themes and Christian theology, particularly tokens " God" and " Him" | our sins were placed on **Him**, and He took the
this was in accordance with **God**'s plan, for **God**
to feel at one with **Him**, to melt into Divinity |
| 81 | Intensifiers (" extremely", " so", " very") paired with adjectives | rm c}$ is **extremely** sensitive to details of the
mood, but she was **so** busy with life that,
, but the tour is **very** much an outdoor one through |

Table 3: Examples of top samples and their contexts that activate latents in regular SAE trained with backpropagation, BP setup. The activating tokens are highlighted in orange, and the color opacity indicates activation strength.

| Latent index | Pattern | The most latent-activating examples (strength descending order) |
|---|---|---|
| 10 | Token "OO" in "GOOGLE" | MAP    G**OO**GLE MAP ) ;
MAP    G**OO**GLE MAP ) )
Island Life (   G**OO**GLE MAP ) ; |
| 16 | Token " resolution" | responding to a mass **resolution** of $1.4
relaxation times obtained from high **resolution** scans
the assay without adding significant **resolution** to the |
| 157 | Tokens " good", " great" | wrong all tournament. As **good** as Celia Ål
sulfur and it wasn't **good**. When we got home
of them talked about how **great** the policy was, an |
| 160 | Tokens " trying", " suggesting", " intended" | size\n\nI am **trying** to change font family and
massive clusters, **suggesting** that discs may have survived
Galafold is not **intended** for concomitant use |
| 487 | The verb "to be" in different forms: " were", " are", rarely " was". | good hiding place. Where **were** you?"\n\n"
back out. "Where **are** you going?"\n\n
in the distance. Where **was** the low hill I remembered |

weight updates are the modified anti-gradients of the $L_2/L_1$ loss (applied to the activations in the bottleneck). The modification is the orthogonalization of $x$ to $w$, or the Gram-Schmidt orthogonalization. This makes the weight updates orthogonal to the weight vector itself. For small learning rate and/or sparsity coefficient it can be thought of as rotating the weight vector away from the input. This results in the weight vector becoming less tuned to the input, and the corresponding neuron activation decaying. The anti-Hebbian update of SoftHebb resembles the anti-gradient of $L_2$ loss on the latents, while the rule from Krotov & Hopfield (2019) to $L_1$ loss. This distinction likely explains the larger effective dictionary size observed in SoftHebb encoders compared to encoders in backpropagation only setups.

Formerly, Elhage et al. (2022) connected vulnerability to adversarial examples to the formation of superposition inside of a network. We believe the adversarial robustness of MLP models trained with SoftHebb is connected to the reduced superposition in features compared to BP training, as seen in SAE bottleneck representations.

## 7 CONCLUSION

We have established a novel theoretical and empirical connection between Hebbian Winner-Take-All (WTA) learning with anti-Hebbian updates and tied-weight sparse autoencoders (SAEs), demonstrating that interpretable, monosemantic features can be learned in LM activation spaces using biologically inspired rules. By framing Hebbian WTA mechanisms

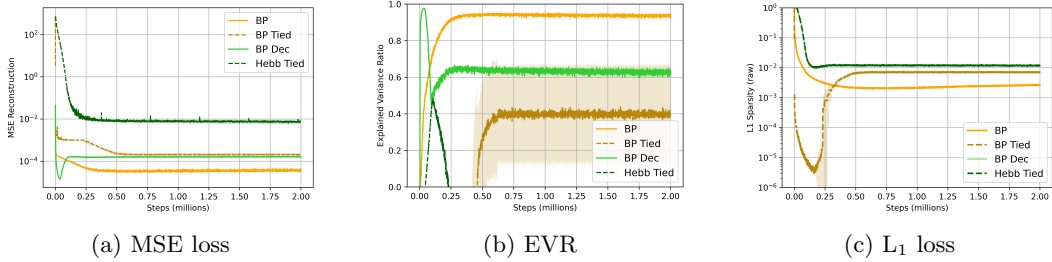

(a) MSE loss          (b) EVR          (c) $L_1$ loss

Figure 1: MSE reconstruction loss, explained variance ratio (EVR), and $L_1$ sparsity during SAE training on `TinyStories-1M` activations. Error bars: mean $\pm$ std over 5 runs. Hebb Tied and BP Dec setups overlap for (c).

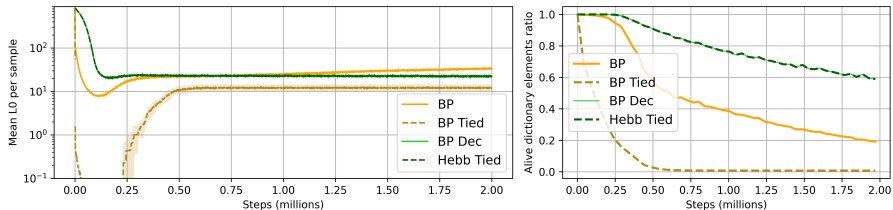

Figure 2: Sparsity statistics: (left) average $L_0$ norm of SAE latents; (right) dictionary size (unique latents that ever activate). Hebb Tied and BP Dec setups overlap.

as an implicit optimization of SAE objectives - reconstruction (MSE) via Hebbian updates and sparsity/orthogonality via anti-Hebbian updates and competition - we bridge the gap between neuroscience-inspired learning and modern mechanistic interpretability research.

## 8 LIMITATIONS AND FUTURE WORK

Although the proposed Hebbian approach underperforms standard backpropagation on direct reconstruction metrics, it offers compelling benefits in terms of biologically inspired mechanisms and potential interpretability. Future work could cover the extension the method to larger architectures and image domain, comprehensive ablation study, and scaling interpretation analysis with autointerpretability score.

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

## A APPENDIX

### A.1 LEARNING RULES

Table 4: Weight update rules for a fully-connected layer: bio-inspired rules and tied weights autoencoder anti-gradient. The bias terms are omitted. For the fast AI implementation of (Krotov & Hopfield, 2018), $i$ is the hyperparameter. diag is a function leaves only elements on the main diagonal, LT() is a function that sets elements above the main diagonal to zero, SM() is the Softmax activation function.

| Method | Weight update |
|---|---|
| Simple Hebbian rule | $\Delta \boldsymbol{W} = \boldsymbol{y}\boldsymbol{x}^T$ |
| Oja's rule | $\Delta \boldsymbol{W} = \boldsymbol{y}\boldsymbol{x}^T - \mathrm{diag}(\boldsymbol{y}\boldsymbol{y}^T)\boldsymbol{W}$ or $\Delta \boldsymbol{w}_k = y_k(\boldsymbol{x} - y_k\boldsymbol{w}_k)$ [4] |
| Sanger's rule | $\Delta \boldsymbol{W} = \boldsymbol{y}\boldsymbol{x}^T - \mathrm{LT}(\boldsymbol{y}\boldsymbol{y}^T)\boldsymbol{W}$ |
| Oja's subspace algorithm | $\Delta \boldsymbol{W} = \boldsymbol{y}\boldsymbol{x}^T - \boldsymbol{y}\boldsymbol{y}^T\boldsymbol{W}$ |
| Tied weight linear AE ($\mathcal{L} = \mathrm{MSE}$) | $\Delta \boldsymbol{W} = \left[\boldsymbol{y}\boldsymbol{x}^T - \boldsymbol{y}\boldsymbol{y}^T\boldsymbol{W}\right] + \left[\boldsymbol{y}(\boldsymbol{x} - \boldsymbol{W}^T\boldsymbol{y})^T\right]$ |
| SoftHebb (Moraitis et al., 2022) | $\Delta \boldsymbol{w}_k = \begin{cases} SM(y_k)(\boldsymbol{x} - y_k\boldsymbol{w}_k), & \text{if } k = \arg\max_k y_k \\ -SM(y_k)(\boldsymbol{x} - y_k\boldsymbol{w}_k), & \text{otherwise} \end{cases}$ |
| Fast AI implementation of (Krotov & Hopfield, 2018) | $\Delta \boldsymbol{w}_k = \begin{cases} (\boldsymbol{x} - y_k\boldsymbol{w}_k), & \text{if } k = \arg\max_k y_k \\ -\Delta(\boldsymbol{x} - y_k\boldsymbol{w}_k), & \text{if } k = i \text{ (hyperparameter)} \\ 0, & \text{otherwise} \end{cases}$ |

### A.2 EXPERIMENTAL DETAILS

During training, we use 32 batches of 512 embeddings, but metrics and losses are averaged over all of them, i.e. all $2^{14}$, so we report the total batch size. For simplicity, we do not initialize the decoder weight vectors in an SAE to be orthonormal and do not normalize them before making a forward pass.

#### A.2.1 HEBBIAN LAYER DETAILS

We use the following hyperparameters of the hebbian layer in our training:

- `layer_type` = HebbSoftKrotovLinear
- `lebesgue_p` = 2
- `weight_distribution` = normal
- `weight_offset` = 0
- `t_invert` ($\tau$) = 50
- `bias` = True
- `lr_bias` = 0.1
- `activation_fn` = exp

The hyperparameters for the learning scheduler in a hebbian layer:

- `lr` = $10^{-2}$

---

[4]Oja's rule is frequently written in the vector form for k-th output layer's neuron ($\Delta \boldsymbol{w}_k = y_k(\boldsymbol{x} - y_k\boldsymbol{w}_k)$) and the scalar form for a weight from i-th input neuron to k-th output neuron ($\Delta w_{ik} = y_k(x_i - y_k w_{ik})$).

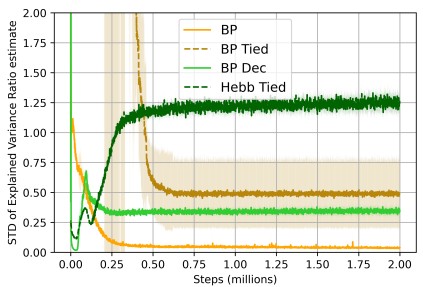

Figure 3: Standard deviation estimate for EVR during training.

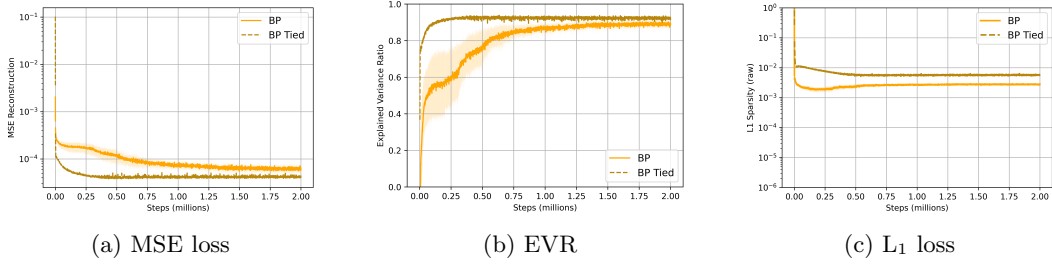

(a) MSE loss          (b) EVR          (c) $L_1$ loss

Figure 4: Ablation of SAE setups to default PyTorch weight initialization scheme. MSE reconstruction loss, explained variance ratio (EVR), and $L_1$ sparsity during SAE training on `TinyStories-1M` activations. Error bars: mean $\pm$ std over 5 runs.

- `adaptive` = True
- `nb_epochs` = 0
- `ratio` = 1
- `speed` = 0
- `div` = 100
- `decay` = constant
- `power_lr` = 0.2

For SoftHebb, the bias term vector of a linear layer is initialized with a constant value $\frac{\tau}{\log \frac{1}{\#neurons}}$. We do not use the learning scheduler, but employ the weight-norm-dependent adaptive learning rate from Journé et al. (2023) for the output neurons in the layer. It means that the learning rate is not the same for all neurons. The bias learning rate is set to $\frac{\texttt{lr\_bias}}{\tau}$. We noticed the original repository not utilizing the `lr_bias` hyperparameter, but rather setting bias learning rate to $\frac{1}{\tau}$, so we modified it.

The change of a k-th bias term in a layer is computed according to the official repository of Journé et al. (2023):

$$\Delta b_k = \frac{y_k - e^{b_k/\tau} \cdot \text{batch\_size}}{e^{b_k/\tau}} \tag{3}$$

The derivation of the update was considered in Moraitis et al. (2022).

A.3  ADDITIONAL EXPERIMENTAL RESULTS

Along with calculating the mean EVR, we analyze standard deviation of the samplewise evaluations, see Figure 3

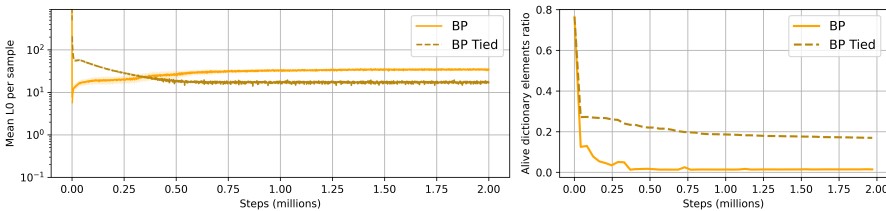

Figure 5: Ablation of SAE setups to default PyTorch weight initialization scheme. Sparsity statistics: (left) average $L_0$ norm of SAE latents; (right) dictionary size (unique latents that ever activate).

### A.4 Ablation of weight initialization

The tied-weight backpropagation SAE setup (BP Tied) shows high variance across runs and hugely underperforms compared to BP setup. We find the reason to be the normal weight initialization adopted from Journé et al. (2023). Ablating the weight initialization of SAE training setups to the default PyTorch scheme, i.e. parametrized Kaiming uniform distribution on the weights and uniform on the bias terms, recovers the performance of BP Tied, see Figure 4 and Figure 5. However, the Hebbian setups degenerated to NaN values, and are not shown on the figures.

