# OpenReview forum: "Hebbian Learning through the Lens of Sparse Autoencoders"
_mathai.club/MathAI/2026/Conference — 2026 Oral_

### Official Review · Reviewer_7fYJ · 2026-03-12
**Hebbian WTA Learning as Implicit SAE Training**

**Rating:** 6
**Confidence:** 3

**Review:**

Interesting theoretical framing connecting Hebbian WTA to SAEs. The anti-gradient interpretation of Hebbian/anti-Hebbian updates is the strongest contribution.
However, the empirical validation is weak to support the claims. In Table 1, BP Dec and Hebb Tied share identical L1, L0, and Dictionary Size values because they use the same frozen Hebbian encoder, so these metrics reflect the shared encoder activations rather than differences in decoder training. More importantly, Hebb Tied’s EVR of −0.52 indicates a serious reconstruction failure, as it performs worse than a mean baseline.
In addition, the experiments are limited to a 1M-parameter toy model (TinyStories-1M)
The idea is promising and worth pursuing, but the paper would benefit from cleaner experimental validation and stronger empirical evidence before acceptance.

---

> ### Author Rebuttal · Authors · 2026-03-13
>
> We thank the reviewer for the careful reading of the paper and for highlighting both the strengths and limitations of our work. We appreciate the positive feedback regarding the theoretical framing.
>
> Below we address the specific concerns.
> 1. The reviewer correctly observes that BP Dec and Hebb Tied share identical, and dictionary size values. This behavior is expected because both methods use the same frozen Hebbian encoder, and therefore produce identical activation patterns. This point is indeed mentioned in the manuscript (line 194) in the discussion of Figure 1c, but we agree that the paper would benefit from stating this more explicitly in the discussion of Table 1 as well. In the final version, we will clarify that.
>
> 2. We agree that the empirical evaluation is currently limited to a 1M-parameter TinyStories model, which should be viewed as a controlled proof-of-concept setting rather than a large-scale empirical validation.This limitation is already discussed in the Limitations and Future Work section.

---

### Official Review · Reviewer_269h · 2026-03-12
**Good biologically inspired paper.**

**Rating:** 7
**Confidence:** 3

**Review:**

This paper establishes a theoretical and empirical connection between Hebbian learning (augmented with anti-Hebbian updates) and tied-weight Sparse Autoencoders (SAEs). The authors propose that Hebbian updates implicitly optimize SAE objectives: reconstruction error  via Hebbian terms and sparsity/orthogonality via anti-Hebbian terms. The method is validated on token embeddings from the TinyStories-1M language model using the SoftHebb rule. The work aims to bridge biologically plausible learning rules with mechanistic interpretability research, specifically addressing the superposition hypothesis.
Pros
-Successfully bridges neuroscience-inspired learning rules (Hebbian) with modern mechanistic interpretability (SAEs), offering a new framework to explain neuronal selectivity.
-Manual inspection confirms that Hebbian SAEs learn monosemantic latents mirroring the benefits of conventional SAEs.
-Provides an argument that anti-Hebbian updates enforce sparsity and feature orthogonality close to explicit L1/L2 penalties in standard SAEs.
-: Demonstrates that interpretable features can be learned using local, gradient-free rules for the encoder, aligning with biological constraints.

Cons
-High variance in Explained Variance Ratio (EVR); some Hebbian runs yield negative EVR, indicating performance worse than predicting the mean.
-Experiments are limited to a small language model (TinyStories-1M); scalability to larger LLMs remains unproven.


The paper makes a significant conceptual contribution by linking Hebbian learning to SAE objectives, offering a fresh perspective on interpretability and biological plausibility. However, the substantial performance gap and stability issues limit its immediate practical utility. The work is theoretically sound but requires stronger empirical validation on larger scales.

---

> ### Author Rebuttal · Authors · 2026-03-13
>
> We thank the reviewer for the careful reading of the paper.
>
> We agree that the empirical evaluation is currently limited to a small language model, which should be viewed as a controlled proof-of-concept setting rather than a large-scale empirical validation. This limitation is already discussed in the Limitations and Future Work section.

---

### Official Review · Reviewer_Yf8t · 2026-03-13
**The article "Hebbian Learning through the Lens of Sparse Autoencoders" investigates the application of SoftHebb for SAEs. The work is structured, original, and links local learning with global optimization. Its significance could be substantial but is currently limited by the small scale of experiments. The theory (anti-gradients) requires refinement, applicability boundaries need clarification, and novelty claims should be more moderate. Overall, it is a worthy contribution at the intersection of neuroscience and AI, needing improvement in the limitations and discussion sections.**

**Rating:** 7
**Confidence:** 4

**Review:**

Review
1.	Assessment of the quality, clarity, originality, and scientific significance of the work
The article presents a study at the intersection of two current areas: research on local learning algorithms (Hebbian learning) and the interpretability of large language models (interpretability via SAE), and addresses the question of biologically plausible algorithms.
• Quality and clarity: The work is written in a structured manner, with clear argumentation. The theoretical part is supported by mathematical derivations (in the appendices). The Introduction and Background section outline the problem and familiarize the reader with the necessary concepts (Hebb's rule, Oja's rule, WTA, SAE, superposition). The experimental setup is described in detail, allowing for reproduction of the results.
• Originality: The use of the biologically plausible SoftHebb rule for training the SAE encoder and the comparison of the interpretability of the resulting latent variables, as well as with different variants of backpropagation, enables the investigation of algorithms alternative to the mainstream on a solid methodological basis and applies the principle that spurred the development of neural algorithms (Hebb, 1949) to a task that is currently typically solved by other methods. The idea that local Hebbian updates can implicitly optimize global objectives of sparse coding is of interest.
• Scientific significance: The work demonstrates that mechanisms used by the brain for learning (local, self-organizing) can lead to the formation of representations similar to those that modern researchers obtain using the computationally expensive backpropagation of error, in the context of searching for alternatives and advancing interpretability ideas. However, the significance is limited by the scale of the experiments (small model, narrow domain).
2.	Main ideas contained in the article
Application of SoftHebb for training SAE: It is demonstrated that Hebbian updates with anti-Hebbian competition mathematically approximate the anti-gradients of the loss function of a sparse autoencoder with tied weights. The authors integrate the SoftHebb rule—a modern version of Hebbian learning with a soft WTA mechanism—into the sparse autoencoder architecture. The encoder is trained using a local rule, while the decoder is either trained via backpropagation (hybrid scheme BP Dec) or is the transposed matrix of the encoder (fully Hebbian scheme Hebb Tied).
Theoretical justification of the connection with SAE optimization: The key theoretical finding of the article is that the authors show: Hebbian (positive) updates in SoftHebb are analogous to anti-gradients for MSE reconstruction, while anti-Hebbian (negative) updates for losing neurons promote sparsity and orthogonality of features. This is not merely a heuristic, but potentially a new perspective on how the same mathematics (optimization of the objective function) can be implemented through different physical mechanisms (global gradient vs. local correlation).
Empirical interpretability: The authors do not limit themselves to metrics (MSE, EVR, L0) but also conduct a qualitative analysis, showing examples of latent variable activations. They demonstrate that the Hebbian SAE (Hebb Tied) finds interpretable patterns (e.g., tokens "edd", "tr", markup tags, religious vocabulary) that surpass in diversity the patterns found by classical BP. However, the extent to which increased diversity enhances interpretability is not addressed in the article.
3.	Degree of novelty, relevance, and significance of the claims
• Novelty: Although the idea of using Hebbian rules for learning has a history of more than 75 years, their application to the task of LM interpretability via SAE is an original contribution. The combination with the current interpretability trend (SAE, dictionary learning) represents a new perspective on Hebbian rules.
• Relevance: Interpretability of LLMs is one of the important problems in modern AI. The search for alternative, more biologically plausible, and possibly more efficient learning methods that lead to monosemanticity is a relevant task.
• Significance of the claims: The attempt to explain why Hebbian learning reduces superposition and therefore may lead to more interpretable features, as well as its connection to robustness against adversarial attacks (reference to Elhage et al., 2022), is potentially important. The hypothesis of potential robustness due to reduced superposition is interesting, but in the work it is not supported by direct experiments on adversarial robustness. The theoretical conclusion that Hebbian rules implicitly optimize the SAE objective is correct within the made assumptions (tied weights, linear approximation), but is not universal. The boundaries of applicability need to be more clearly delineated.
4.	Assessment of compliance with requirements for logic of presentation, formatting, terminology
• Logic of presentation: The logic is well-structured: from the problem (non-biological nature of BP) and goal (using Hebb) through the method (integration of SoftHebb into SAE) to experiments and conclusions. The experiments logically compare 4 configurations, allowing assessment of each component's contribution. However, in Section 3.2 (theoretical analysis), explanations for the used transformations are lacking.
• Terminology: The terminology is used correctly. The authors carefully distinguish concepts: Hebbian learning, anti-Hebbian, Oja's rule, SoftHebb, WTA. Terms from the interpretability field are used correctly (superposition, monosemanticity, latent variables).
5.	Substantiated remarks, recommendations for revisions, supplementation of material
Section 6 (Discussion) requires refinement: Item on anti-Hebbian updates: The statement "The anti-Hebbian update of SoftHebb resembles the anti-gradient of L2 losses on latent variables, whereas the rule from Krotov & Hopfield (2019) is the anti-gradient of L1 losses" is key, but it is too brief. It is recommended to move the mathematical justification of this difference to the main body of the article (or to the Appendix) and present it more rigorously, possibly with brief derivations; Item on superposition and robustness: The connection between reduced superposition in Hebb Tied and potential robustness to adversarial attacks is a very interesting assumption. However, in its current form, it appears as an offhand remark at the end of the section. It should either be supported by a reference to a separate experiment (even a small one) or moved to the "Future Work" section as a hypothesis for verification.
Practical significance: In the "Limitations" section, the authors mention that their approach underperforms BP on reconstruction metrics. It would be useful to add a point on whether the effort is worthwhile. Yes, the metrics are worse, but perhaps Hebbian ensembles possess other valuable properties (e.g., require orders of magnitude less data for training, or do not require large GPU clusters due to the locality of updates). If such data is unavailable, it should be explicitly stated that this is currently a proof-of-concept demonstrating the fundamental possibility, not a production-ready method.
Regarding the assessment of pioneering novelty: The claims about novelty are generally justified but require more precise positioning relative to prior work and more cautious formulations concerning unconfirmed hypotheses.
Expand the discussion of limitations: Explicitly state that the theoretical equivalence is proven for the linear case and tied weights; nonlinear architectures and untied weights require separate investigation.
Conclusion:
The article presents a well-argued study at the intersection of neuroscience, optimization, and AI interpretability. The theoretical connection between Hebbian rules and the objective functions of sparse autoencoders can make a substantive contribution to bridging neuroscience and machine learning. The main remarks concern insufficient development of the theoretical part (discussion of anti-gradients) and the part regarding the method's limitations. Additionally, the degree of novelty of the work's results is formulated too boldly.

---

> ### Author Rebuttal · Authors · 2026-03-13
>
> We thank the reviewer for sending his auspicious review.

---

### Decision · Program_Chairs · 2026-03-14

**Decision:**

Accept (Oral)

**Comment:**

Dear Author(s),

On behalf of the Program Committee of the International Conference on Mathematics of Artificial Intelligence (MathAI 2026), we are pleased to inform you that your paper has been accepted for an oral presentation at MathAI 2026.

Your paper was evaluated through a rigorous two-stage review process involving both automated screening and expert review by members of the Program Committee. The reviewers recognized the quality and contribution of your work.

Presentation details:

- Format: Oral presentation (15–20 minutes + 5 minutes Q&A)
- Mode: You may present either in person (offline) at the conference venue in Sirius, Russia, or remotely via Zoom. Please indicate your preferred mode when confirming your participation.
- Conference dates: Marh 30 - April 3, 2026
- Website: https://mathai.club

Next steps:

1. Please confirm your participation and presentation mode by replying to this email mathai.club@yandex.ru no later than March 15, 2026 18:00 Moscow time.
2. If you plan to attend in person, the organizing committee will provide accommodation details separately.
3. Please prepare your final camera-ready manuscript according to the formatting guidelines available at https://mathai.club and upload it to OpenReview by March 15, 2026 18:00 Moscow time.

Should you have any questions regarding the program, logistics, or your presentation slot, please do not hesitate to contact us.

We look forward to your contribution to MathAI 2026.

With kind regards,

MathAI 2026 Program Committee
International Conference on Mathematics of Artificial Intelligence
https://mathai.club
OpenReview: https://openreview.net/group?id=mathai.club/MathAI/2026/Conference
Telegram: https://t.me/MathAI_club
Email: mathai.club@yandex.ru